# The New Frontier of Immunotherapy: Chimeric Antigen Receptor T (CAR-T) Cell and Macrophage (CAR-M) Therapy against Breast Cancer

**DOI:** 10.3390/cancers15051597

**Published:** 2023-03-04

**Authors:** Giuseppe Schepisi, Caterina Gianni, Michela Palleschi, Sara Bleve, Chiara Casadei, Cristian Lolli, Laura Ridolfi, Giovanni Martinelli, Ugo De Giorgi

**Affiliations:** IRCCS Istituto Romagnolo per lo Studio dei Tumori (IRST) “Dino Amadori”, 47014 Meldola, Italy

**Keywords:** chimeric antigen receptor (CAR), macrophages, T cells, immunotherapy, breast cancer

## Abstract

**Simple Summary:**

To date, different therapeutic strategies, including immunotherapies, have been shown to prolong survival in breast cancer patients, representing one of the most common malignancies. Our article deals with chimeric antigen receptor-based immunotherapy in breast cancer.

**Abstract:**

Breast cancer represents one of the most common tumor histologies. To date, based on the specific histotype, different therapeutic strategies, including immunotherapies, capable of prolonging survival are used. More recently, the astonishing results that were obtained from CAR-T cell therapy in haematological neoplasms led to the application of this new therapeutic strategy in solid tumors as well. Our article will deal with chimeric antigen receptor-based immunotherapy (CAR-T cell and CAR-M therapy) in breast cancer.

## 1. Introduction

Breast cancer (BC) is the most common female cancer worldwide. According to Globocan, it is the number one diagnosed cancer with an estimated 2.3 million new cases (11.7%) globally in 2020, and is the fifth leading cause of cancer mortality [1]. The BC incidence is increasing, especially in highly developed countries where screening strategies help to reduce cancer mortality while in poor-developing countries, the BC incidence remains low but the mortality rate is still higher [1]. However, in advanced countries, the diagnosis of de novo metastatic BC still represents approximately 3% to 6% of new BC diagnoses and has not declined despite the wide diffusion of mammography screening [2].

BC is a very heterogeneous disease, clinically distinguished into several subtypes according to the expression of hormone receptors (HRs) and the human epidermal growth factor receptor 2 (HER-2) status: luminal BC, HER-2 positive BC, and triple-negative BC (TNBC). HR and HER-2 are the targets for numerous specific treatments in early and advanced settings. TNBC is defined by the lack of expression of HR and HER-2 and accounts for approximately 15% of all BCs [3].

TNBC has long been considered a major unmet need due to its aggressive behavior and poor prognosis related to its deficiency of specific therapeutic targets. These tumors tend to relapse early and rapidly metastasize in the lungs, liver, and central nervous system, determining a worse survival [4]. For these reasons, chemotherapy is still a cornerstone in the treatment of this BC subtype.

Immunotherapy with checkpoint inhibitors was shown to be effective in TNBC, especially in the case of programmed death 1 (PD-1) expression in tumor tissue [5]. Furthermore, TNBC has the highest tumor mutational burden (TMB) among all BC subtypes [6]. A high mutational level can lead to the production of tumor “neoantigens” which could be recognized by antigen-presenting cells in the tumoral microenvironment (TME) enhancing antitumor immune response [6]. Even if BC has always been considered a poorly immunogenic tumor, TN and HER2+ subtypes show considerable immune infiltration. As a demonstration, tumor-infiltrating lymphocytes (TILs) are frequently present in TN and HER2+ tumor samples and are associated with good prognosis and are predictive of immunotherapy efficacy [7,8,9]. Otherwise high immune infiltration has a completely different effect in the luminal subtypes and lobular BC, suggesting a bad prognosis [5,10,11,12]. Especially for TNBC, the intrinsic molecular characteristics (determined by mRNA profiles, gene expression, and proteomics) can distinguish different intrinsic subtypes of TNBCs defined as basal-like 1 or 2, luminal androgen receptor, and mesenchymal tumors [13]. Each intrinsic subtype is associated with an individual TME, shaped by the molecular features and genomic signatures of cancer cells [14]. The quality of immune cells and distribution in the tumoral tissue is also important in TNBC, distinguishing between “cold tumors” and “inflamed tumors” and inflammation areas in the stromal tissue, margins, or fully inflamed tumoral tissue [15,16]. Generally, immune-rich early TNBCs have less clonal heterogeneity, somatic mutation, and a minor expression of neoantigens but a high expression of TILs, CD8 + T cells, or memory T cells [17]. Conversely, metastatic sites seem more heterogeneous and immunodepleted with fewer TILs, CD8 + T cells, or dendritic cells, a low TMB, and increased clonal diversity [18,19]. In the metastatic environment, there is instead a greater presence of metastasis-associated macrophages (MAMs) with a pro-tumoral phenotype that is able to increase immune escape strategies and cancer diffusion [20]. Tumor-mediated immune suppression is a real issue responsible for acquired resistance to active immunotherapy (ICIs) [21].

Targeting the immune system with a combination of different targets, especially in advanced BC, will become a valuable therapeutic strategy to achieve the best survival results. This strategy aims to convert non-responders to responder patients, maintain an achievable lasting response and overcome acquired immune resistance. Interfering with immune evasion, promoting the antitumor phenotype of immune cells, or enhancing antitumor immunity are the expected goals [22]. Many early-phase clinical trials are ongoing in several solid tumors (including BC patients) with new active compounds targeting macrophages or neutrophils [23]. New emerging treatments in solid tumors are now being used in immunotherapy after the incredible results that have been achieved in the onco-hematology field. Among these is adoptive cell therapy (ACT), that exploits TILs or T cells genetically engineered to express modified T-cell receptors (TCR) or chimeric antigen receptors (CAR).

CAR-based therapies with T cells or natural killer (NK) cells are promising as potential practice-changing effectors in BC, especially for tumors with poor targetable antigens (like TNBC), even if there are still significant limitations depending on the resistance of an unfavorable TME and side effects [24,25]. The presence of an extracellular matrix (ECM) in the tumor stroma constitutes a physical barrier to the transfer of CAR-T cells. Macrophages engineered with CAR (CAR-M) may overcome this barrier by producing metalloproteinases and can enhance the antitumor effect thanks to antigen-specific phagocytosis [26,27]. In our review, we give an overview of the potentiality of CAR-based therapies in BC.

## 2. CAR Molecule Structure

A CAR is an artificial fusion protein that is composed of an extracellular antigen binding domain which includes an antigen recognition domain, for example, a mAb-derived single chain variable fragment (scFv) that is involved in the binding between the T cell and a tumor-associated antigen (TAA) [28]. A hinge region is linked with the scFv, providing CAR flexibility; its length can be modified to optimize the distance between CAR-T cells and targeted cancer cells and ameliorate the signal transduction process [29]. Moreover, a transmembrane domain is involved in intracellular signal transmission pathways. For this purpose, this region includes both costimulatory and signaling domains (e.g., CD3ζ, also called CD247) responsible for CAR-T cell activation [28].

## 3. The CAR-T Cell Generations

Ameliorating CAR vectors can improve the safety and efficacy of CAR-T-cell therapy [30]. For this purpose, several CAR generations have been conceived, and currently, the fifth generation is already being tested [31,32]. The principal differences among the CAR generations consist of specific costimulatory molecules. The first generation contains only the CD3ζ signaling end domain, whose linking with the extracellular scFv modifies and activates T cells [33]. However, due to its short survival time and incomplete T-cell activation, it was necessary to conceive a second and third generation of CARs, which include one or two additional costimulatory molecules (respectively), such as CD27, CD28, 41BB, ICOS, and OX-40. These molecules increase the cells’ persistence and cytolytic capacity [34,35,36]. T cells redirected for universal cytokine-mediated killing (TRUCKs) or “armored CAR-Ts” represent the fourth generation of CARs, which includes a nuclear factor of the activated T cells (NFAT) domain [37]. The domain promotes cytokine secretion, mainly interleukin (IL)-12, IL-15, and the granulocyte–macrophage colony-stimulating factor (GM-CSF), which aims at modulating the anti-tumor microenvironment. In fact, armored CAR-Ts carry out a simultaneous antitumor activity directed to both tumor cells expressing and not expressing the CAR-targeting antigen. The other advantage of this strategy is determined by the local release of IL-12, with a lower risk of systemic toxicity related to the cytokine secretion [38]. Such CAR-Ts can be tested for targeting TNBC-related antigens.

In recent years, a fifth generation of CARs has been developed. It contains a fragment of IL-2 receptor β (IL-2Rβ), which induces the secretion of Janus kinases (JAKs) and signal transducer and activator of transcription (STAT)-3/5 [31,39]. Such novel CAR generation aims to avoid terminal phenotypic differentiation of effector cells; consequently, fifth generation CAR is able to promote their expansion in vitro, and their persistent cytotoxicity in vivo [40].

## 4. Targets for CAR-T Cell Therapy in BC

The development of CARs has led to searching for new targets for cancer therapy, especially for histologies without ERBB2 and HR expression [24], such as TNBC (Figure 1). All studies testing potential targets for CAR-T cell therapy in BC tumors are shown in Table 1.

Abbreviations: CAR: chimeric antigen receptor; CEA: carcinoembryonic antigen; EpCAM: epithelial cell adhesion molecule; FR: folate receptor; HER2: human epidermal growth factor receptor 2; MUC1: Mucin1; PRLR: prolactin receptor; ROR1: receptor tyrosine kinase-like orphan receptor; TEM8: tumor endothelial marker 8; TNBC: triple-negative breast cancer; VEGFR1: vascular endothelial growth factor receptor 1.

### 4.1. Integrins

In our research, we paid attention to the membrane receptors to identify a specific target for CAR molecules. In this context, Integrins represent a potential target for CARs because of their proven involvement in cell proliferation and metastatization and because of their high expression in BC [41]. In particular, CAR molecules targeting αvβ3-integrin were conceived and tested, showing their cytolytic activity against different tumors in vitro, including MDA-MB-231 TNBC cell lines. After testing these molecules in vivo, some complete responses were reported in mice that were affected by metastatic melanoma [43]. Moreover, as reported by some studies, this therapy demonstrated selective cytotoxicity against αvβ3-expressing cell lines without involving normal cells [42,43,44]. Therefore, based on what has been reported, αvβ3CAR-T cell therapy seems promising and deserves further study to verify its efficacy in BCs.

### 4.2. Mesothelin

Another potential target for CAR molecule development is Mesothelin, a tumor differentiation glycoprotein that is involved in cell adhesion, which is normally expressed on the mesothelial cells but is overexpressed in several solid neoplasms including TNBC [64,65]. Its activity in oncogenesis through different cell signaling pathways such as MAPK, PI3K, and NF-kB has been reported [83]. Since its overexpression has been reported in 67% of TNBCs, with a limited expression in normal breast cells [84], Mesothelin represents an appealing target for CAR molecule development. In this regard, Hu et al., evaluating the Mesothelin expression on three TNBC cell lines, such as MDA-MB-231, BT-549, and Hs578T, reported that only BT-549 cells expressed the molecule. Then, the authors produced second-generation mesothelin-redirected CAR-Ts and tested it in vitro and in vivo. It is noteworthy that the researchers disrupted the gene locus of PD-1 in T cells before CAR transgene insertion. Their CAR-Ts demonstrated an interesting increase in antitumor activity and cytokine secretion against PD-L1-expressing tumor cells in culture [85]. This is probably due to the high expression of PD-L1 in TNBC cells [86], suggesting a potential use of these CAR-Ts in order to overcome the suppressive effects of PD-1/PD-L1 axis in BCs [85]. Thanks to these interesting data, some clinical trials have been developed and are currently underway, with the aim of evaluating the activity of CAR molecules in TNBCs. In particular, a Phase I clinical trial (NCT02792114) is evaluating the safety and tolerability of Mesothelin-redirected CAR-Ts in metastatic/advanced Mesothelin-expressing BCs, including TNBCs. Another Phase I/II clinical trial (NCT02414269) is testing second generation Mesothelin-redirected CAR-Ts in different tumors, including BC. Moreover, two other clinical studies (NCT02580747 and NCT01355965, the latter only in Mesothelioma) have been completed but no official data have been published yet.

### 4.3. TEM8

Some studies have demonstrated that the endothelium of several neoplasms often overexpresses an integrin-like protein called tumor endothelial marker 8 (TEM8), also known as ANTXR1, and is usually expressed during endothelial cell development but rarely in adults [47,48]. As evidence of this, an elevated expression of TEM8 was found both in invasive/relapsed BC [49,50] and in several BC cell lines [51], so the upregulation of this molecule could represent a potential target for CAR-T cell development [87,88]. In this regard, a single dose of a specific L2CAR-T cell therapy, derived from the L2 Monoclonal Antibody (Mab) against TEM8, showed a complete response against TNBC in vitro and significant cancer reduction in vivo and TNBC xenografts [51]. For these reasons, TEM8 represents a promising target for CAR-T cell therapy against TNBC.

### 4.4. MUC1

In TNBC, MUC1 represents a highly selective overexpressed target [66]. It is a glycosylated transmembrane molecule with altered epithelia [66]. It produces mucin, which protects cells against pathogens [67,68,69]. Tumor cells overexpress MUC1, activating the intracellular pathways involved in cancer proliferation [66,69]. Jiang et al.’s recent cohort study, involving more than 5800 BC patients, demonstrated the predictive role of MUC1 and its correlation with a poor prognosis [89]. In particular, neoplasms overexpress a hypo-glycosylated variant of MUC1, also known as tumor-specific MUC1 (tMUC1) which exposes new epitopes for the immune system [90]. For this purpose, antibodies specifically binding to tMUC1 have been developed and tested [91]. One of these molecules, the so-called TAB004, served as a reference point for creating a CAR molecule containing its extracellular scFv. The derived CAR-T cells, known as MUC28ζ CAR-T cells demonstrated their efficacy in heightening the expression of both leukocyte activation markers and cytokines in vitro. These effects led to significant cell lysis in vitro and a reduction in cancer cell growth in vivo [66]. Recently, a new CAR molecule targeting tMUC1, known as huMNC2-CAR44, has been activated in a clinical trial recruiting 69 (HER2-positive, HER2-negative, triple-negative) BC patients; the estimated study completion date is 15 January 2035 (NCT04020575). Another Phase I study is currently testing the safety, tolerability, feasibility, and preliminary efficacy of the administration of CAR-T cells targeting tMUC1 in 112 patients with advanced tMUC1-positive solid tumors (including BC) and multiple myeloma. The estimated study completion date is 31 October 2036 (NCT04025216).

### 4.5. ROR1

Receptor tyrosine kinase-like orphan receptor (ROR)1 is a highly expressed molecule during embryogenesis but not in adults. BC cells highly express ROR1, especially in cases with a poor prognosis; ROR1 overexpression was found in some TNBC cell lines (e.g., in MDA-MB-231) but not others [70]. ROR1-based CAR-T cell therapy was also shown to induce MDA-MB-231 apoptosis in tumor models through significant IL-2 and IFNγ production [92]. A Phase I trial is testing the ROR1-specific CAR-T cells’ efficacy in 60 subjects with hematological and solid tumors, including triple negative BC. Patients will be followed up for approximately 15 years after study completion. The estimated study completion date is 1 December 2023 (NCT02706392). The first study results were recently published, suggesting a better CAR-T cell therapy efficacy from adding Oxaliplatin to the lymphodepletion regimen [93]. Another Chinese Phase I study is currently recruiting 40 patients with advanced solid tumors (including BC) to investigate the efficacy of TILs and CAR-TILs against several molecular targets, including ROR1, MUC1, HER-2, Mesothelin, PSCA, EGFR, GD1, GPC3, Lewis-Y, AXL, Claudin18.2/6, and B7-H3. The estimated study completion date is 1 January 2035 (NCT04842812).

### 4.6. Natural Killer Group 2, Member D Ligand (NKG2DL)

Under certain pathological conditions, both innate and adaptive immune cells (including CD8+ and some CD4+ T cells, NK cells, γδ T cells) express a type II transmembrane protein, called natural killer group 2, member D (NKG2D) [74], which in turn contribute to enhancing cytotoxicity and production of cytokines by effector cells and promoting their proliferation and survival. Moreover, NKG2D can cooperate with other receptors (including TCR in T cells or NKp46 in NK cells) by acting as a costimulator for their responses [94]. In tumor cells, including TNBC cells, an upregulation of stress-induced ligands has frequently been reported; NKG2D can naturally recognize these ligands [95] so it has been considered as a potential target for immunotherapy in several studies. CAR molecules that were obtained by fusion of the full-length NKG2D with the CD3z cytoplasmic region together with endogenous DAP10 costimulation, were demonstrated to react with NKG2DL-expressing tumor cells through cytokine and chemokine production, thus enhancing cytotoxicity [96]. These results were also confirmed by in vivo studies [97,98]. More recently, NKG2DL-redirected CAR-Ts were tested by Han et al. in TNBC cell lines and TNBC mouse models [99]. In this case, NKG2DL-redirected CAR-Ts were obtained by the fusion between the extracellular domain of human NKG2D and the TCR CD3z alone or co-stimulatory domains, such as 4-1BB or CD27. The authors demonstrated that the elevated expression of CD25 and the presence of IL-2 were required to promote CAR-T expansion in vitro in the absence of any costimulatory domains. Moreover, NKG2DL-redirected CAR-Ts were able to recognize and kill TNBC NKG2DL-expressing MDA-MB-231 and MDA-MB-468 cell lines [99]. Based on these results, a Phase I clinical trial (NCT04107142) evaluated the safety and tolerability of NKG2DL-redirected CAR-T cells in patients with various solid neoplasms including TNBC, but, to date, no results have been reported yet.

### 4.7. Chondroitin Sulfate Proteoglycan 4 (CSPG4)

CSPG4 is a hyperglycosylated transmembrane protein with a low expression in normal tissues and hyperexpressed in several tumor types, including TNBC. It has been suggested that CSPG4 is involved in the neuronal network regulation and epidermal stem cells homeostasis [55]. Second-generation CSPG4-redirected CAR-Ts were tested in various CSPG4-expressing cell lines (including SENMA, UACC-812, CLB, MDA-MB-231, MILL, PHI, and PCI-30), and demonstrated a significant capacity in cell growth suppression [56]. The same results were obtained in preclinical mouse models of several human tumors (including BCs). In another study, second-generation CSPG4-redirected CAR-Ts using murine-based scFvs reported target antigen-dependent cytotoxicity and cytokine secretion against several tumor (including BC) cell lines [100].

### 4.8. EpCAM

Epithelial cell adhesion molecule (EpCAM) represents a well-known molecule whose expression has been related to poor prognosis and tumor metastatization [71]. Several treatment strategies targeting EpCAM have shown benefits for different tumor types. Currently, a Chinese clinical trial is recruiting patients with nasopharynx neoplasms and BC, which aims to evaluate the safety of the engineered CAR-T cells recognizing EpCAM. These molecules were developed through lentiviral transduction of the third generation of CAR genes. Different patient cohorts will receive the experimental treatment in a dose-escalating manner; the estimated study completion date was set for July 2022 (NCT02915445).

### 4.9. Intercellular Adhesion Molecule-1 (ICAM-1)

ICAM-1 is a transmembrane protein that is involved in white blood cell diapedesis. It is overexpressed on the surface of many cancer cells, including TNBC cells [60]. ICAM-1 plays a role in tumor growth, invasion, and metastasis [61]. In order to avoid CAR-T-related cytotoxicity in normal cells, Park et al. generated CAR-Ts with micromolar (instead of nanomolar) affinity, and demonstrated that these ICAM-1-redirected CAR-Ts were more efficacious and safe than their higher affinity homologs [62]. More recently, the same results were confirmed in preclinical models [63].

### 4.10. HER-2

BC with HER2 overexpression represents the tumor subgroup for which CAR-T cells have been designed [101]. Several clinical trials are currently ongoing to test CAR molecules targeting HER-2. One of them is a Phase I trial, testing the safety and preliminary therapeutic efficacy of CCT303-406 cells in 15 patients with HER-2-positive stage IV solid tumors (that have failed standard treatment of relapsed or difficult-to-treat), including BC. The estimated study completion date is April 1, 2023 (NCT04511871). An American, multicenter, Phase I/II trial is currently ongoing, recruiting 220 patients with HER-2-positive tumors, including BC, to evaluate the safety, tolerability, and clinical activity of HER2-specific dual-switch CAR-T cells, BPX-603, administered with rimiducid. The estimated study completion date is January 2, 2025 (NCT04650451). Another American dose-escalation study is being conducted at the City of Hope Medical Center (Duarte, CA) to investigate the side effects and the best dose of HER2-CAR-T cells in treating patients with BC metastasized to the brain or leptomeninges. For this purpose, 39 patients will receive HER2-CAR-T cells via intraventricular administration over five minutes once weekly for three doses, which could be implemented at the principal investigator’s discretion. Patients will be followed up at the end of treatment at 4 weeks, 3, 6, 8, 10, and 12 months, and then for up to 15 years. The estimated study completion date is 31 August 2023 (NCT03696030). A third American Phase I trial is currently studying the safety and efficacy of combining HER2-specific CAR-T cells with an intra-tumor injection of CAdVEC, an oncolytic adenovirus designed to help the immune system activation against cancer. Our trial is recruiting 45 patients with HER-2 positive tumors. The estimated study completion date is 30 December 2038 (NCT03740256). However, although most patients did not have significant complications. In some cases, the non-specificity of HER-2 expression between tumor cells and healthy cells can lead to serious side effects; a case of cardiopulmonary failure from excessive T-cell activation has been reported [102].

Therefore, to avoid drawbacks related to the non-tumor-specificity of the marker, the research sought to deepen its understanding of the receptor to evaluate whether it was possible to find a more specific variant in BC. In this regard, a potential antigen is p95HER2, a truncated version of HER2 was found in 40% of HER2-positive BCs. This variant is more tumor-specific than the constitutive form since it was not found in normal breast cells. p95HER2 has already been evaluated as a target for a bispecific antibody against cancer cells in vitro and in vivo without significant side effects. Due to the encouraging results reported, this variant could, therefore, be a future target for developing CAR-based therapies [103,104].

### 4.11. VEGF

Vascular endothelial growth factor receptor (VEGFR)1 is a Tyrosin-kinase receptor that is involved in the migration and survival of hematopoietic stem cells, and its overexpression is related to the process of BC metastatization [105,106]. Therefore, VEGFR1 represents a potential candidate for immunotherapy. To date, VEGFR1 was tested as a part of a VEGFR1-CD3 bispecific antibody and demonstrated promising results against MDA-MB-231 and MDA-MB-435 TNBC cell lines. These results warrant further studies on VEGFR1 activity, for example as a target for CAR-based therapies [107]. Moreover, because the main functions of the normal endothelial cells are VEGFR2-dependent [105], VEGFR1 inhibition could prevent endothelial toxicity. At present, this is only a hypothesis, so further investigation is needed.

### 4.12. c-MET

Hepatocyte growth factor receptor, also called c-Met, is a cell-membrane protein tyrosine kinase that is expressed in several types of solid neoplasms, including BC [108]. Onartuzumab, an anti-c-Met monoclonal antibody, has been administered in patients with metastatic, solid tumors [109,110,111,112]. Tchou et al. tested c-Met as a potential target for CAR-T cell therapy; for this purpose, the scFv of the CD19 binding domain of a CD19-CAR molecule was substituted for that of onartuzumab, and then its effectiveness against BC cells was confirmed in vitro and in vivo [113]. Subsequently, the new c-Met CAR-T cells were administered through a single intratumoral mRNA injection in a BC patient cohort (NCT01837602). The injections were well tolerated, and no significant drug–related adverse events were reported. Moreover, analyzing tumor specimens (four TNBC and two ER+ HER2-negative BCs) in which the CAR-T cells were injected, there was wide tumor necrosis at the injection site and macrophage infiltrates within the necrotic areas [113].

### 4.13. AXL

AXL, a receptor of the TAM tyrosine kinase receptor family, and its high-affinity ligand, called growth arrest-specific protein 6 (GAS6), are involved in cancer cell expansion, metastasization, and survival; moreover, AXL low expression in adult normal cells and its overexpression in several tumor types (including BC) and some cell lines (including MDA-MB-231 [114]) make AXL a potential target for CAR molecule development [52,115,116].

Wei et al. developed AXL-redirected CAR-Ts using an AXL-specific scFv; these CAR-Ts were tested in AXL-expressing TNBC cell line MDA-MB-231, and demonstrated an antigen-dependent cytotoxicity and cytokine production; these results were confirmed in an in vivo evaluation in MDA-MB-231-established xenograft models [53]. Other researchers generated AXL-redirected CAR-Ts with a constitutively activated IL-7 receptor (C7R); they demonstrated a significant tumor cell killing capacity, which was more efficacious than by using conventional AXL-redirected CAR-Ts, in TNBC MDA-MB-231 and MDA-MB-468 cell lines. This improvement was probably due to the co-expression of C7R, which helped to prolong survival and reduce rates of tumor relapse [54]. However, further investigations are required to confirm these results.

### 4.14. Disialoganglioside GD2

GD2 is a surface protein that is normally expressed only in peripheral nociceptors, neurons, and melanocytes; consequently, GD2 expression has been demonstrated in neuroectoderm-derived tumors, such as melanomas and neuroblasomas [117]. Its cell-type restriction render GD2 a potential target for CAR molecule development. In fact, GD2 has been studied mainly as a target for treatments against neuroblastoma [118]. However, more recently, Seitz et al. used the scFv derived from anti-GD2 mAb dinutuximab to produce GD2-redirected CAR-Ts. The researchers evaluated the GD2 expression in several TNBC cell lines, demonstrating a very low expression in MDA-MB-231, whereas Hs578T and BT-549 uniformly expressed GD2. However, in an in vitro assay, these CAR-Ts did not demonstrate any specific tumor cell killing activity towards MDA-MB-231, whereas they induced specific cytotoxicity and cytokine production upon co-cultivation with the Hs578T and BT-549 cell lines [59]. To date, one study is currently testing the feasibility, safety, and efficacy of multiple fourth generation CAR-T cells targeting Her2, GD2, and CD44v6 surface antigen in BC (NCT04430595), but no results have been published yet.

### 4.15. PRLR

In mammals, prolactin is an important hormone for milk secretion and mammary tissue growth by binding with the prolactin receptor (PRLR) in the breast glands [119]. PRLR is overexpressed in some BC histotypes, especially in the MDA-MB-231 TNBC cell line and even more in T47DHER2+ and SKBR-3 cell lines [120]. This correlation between PRLR and HER2 expression could led to the development of a CAR-based therapy targeting PRLR against BC. However, in a combination of these two targets, a bispecific antibody cytotoxicity was reported, mainly caused by the combinatory inhibition of the two rather than the effect of T-mediated cytotoxicity [121]. Although the results seem promising, future studies on CAR-based therapies targeting PRLR must avoid causing cross-toxicity of other organs expressing the same receptor, such as the prostate, liver, etc. [122].

### 4.16. CEA

CEA is a well-known tumor marker that is expressed in several solid neoplasms [75]. In normal cells, only a small amount of CEA is expressed in the cell membrane, especially toward the cell cavity, under physiological conditions to avoid recognition by CAR-T cells targeting CEA. To date, a Phase I-II study is recruiting 40 patients with different solid tumors to test the efficacy and safety, recommended dose, and infusion plan of CEA-targeted CAR-T cells therapy. The estimated study completion date is 30 April 2023 (NCT04348643).

### 4.17. CD44v6

CD44 variant domain 6 (CD44v6), a CD44 family member, has demonstrated its role in tumorigenesis, tumor cell invasion, and metastasis. In normal tissue, its presence is reported only on epithelial and hematopoietic cell subgroups, especially during embryogenesis and hematopoiesis [76]. In contrast, it is expressed in multiple squamous cell carcinomas, in a proportion of adenocarcinomas of differing origin, a proportion of lymphoma and melanoma, so it represents an attractive target for cancer therapy. A Phase I-II clinical trial is currently investigating the feasibility, safety, and efficacy of CD44v6 CAR-T cell therapy in 100 patients with several tumors, including BC. Understanding more about 4SCAR-CD44v6 T cell functions is another objective of this trial. The estimated study completion date is 31 December 2023 (NCT04427449).

### 4.18. Trophoblast Cell-Surface Antigen 2 (TROP2)

TROP2 is a transmembrane protein that is expressed on human trophoblast cell surface and is often present in several epithelial tumor types (including TNBC), in which it is associated with poor prognosis [57]. Zhao et al. developed and tested in vitro (against gastric cancer cell lines) and in vivo bispecific TROP2- and PD-L1-redirected CAR-Ts [58]. The authors reported a higher antitumor activity with bispecific CAR-Ts compared with monospecific CAR-Ts. In spite of CAR-T-mediated TROP2 targeting in TNBC not being comprehensively investigated, the results obtained in gastric cancer cell lines warrant further investigation in other tumor types, including TNBC.

### 4.19. Epidermal Growth Factor Receptor (EGFR)

EGFR is a transmembrane glycoprotein belonging to the ERBB receptor tyrosine kinase family, that is involved in tumor growth and metastatization [72]. Its overexpression has been reported in several tumor types, including TNBC. Li et al. produced EGFR-redirected CAR-Ts using the non-viral piggy Bac transposon system, and demonstrated their antitumor activity firstly in human lung tumor xenografts and then in a phase I clinical trial (NCT03182816) against non-small cell lung tumors [123,124]. With regards to TNBCs, Liu et al. tested EGFR-redirected CAR-T antitumor activity in vitro and in vivo, reporting EGFR overexpression in several tumor cell lines, such as Hs578T, MDA-MB-468, and MDA-MB-231, respectively. These CAR-Ts demonstrated antitumor activity and cytokine secretion in these cell lines [125] Recently, Xia et al. generated third-generation EGFR-redirected CAR-Ts, which demonstrated antitumor activity, specific cytokine production. Moreover, the authors found that T cell activation markers (such as CD25 and CD69) were upregulated in case of co-cultivation with EGFR-positive TNBC cell lines [126]. However, more preclinical and clinical studies are needed to confirm these findings. Several clinical trials (NCT03182816, NCT02873390, NCT02862028, and NCT03170141), are testing the effects of EGFR-redirected CAR-Ts in regards to the production of anti-CTLA-4, anti-PD-1, or anti-PD-L1 antibodies in EGFR-positive solid tumors [127]. With regards to TNBC, two studies are ongoing. The first one (NCT05341492) is evaluating the safety and efficacy of EGFR/B7H3 CAR-T cell therapy in EGFR/B7H3-positive advanced solid cancers (including TNBC), and the second one is testing the anti-tumor activities and safety profiles of CAR-EGFR-TGFβR-KO T cell therapy in previously treated advanced EGFR positive solid tumors (including TNBC) However, no results regarding TNBC patients have yet been officially reported

### 4.20. Prostate-Specific Membrane Antigen (PSMA)

Although at first glance it does not appear to be related to BC, it has recently been demonstrated that this molecule is present in circulating breast cancer cells, and is related to a worse prognosis [77]. Although PSMA is expressed in normal prostate and is upregulated in prostate tumors, it is not prostate cancer restricted. In TNBC, PSMA is currently under evaluation as a target for CAR molecule development. In an ongoing, open label, Phase I trial, an anti-PSMA/CD70 bi-specific CAR-T cell therapy has been tested in several cancer types (including TNBC) expressing PSMA or CD70, another potential tumor target, that is overexpressed in many cancer types and scarcely expressed in normal tissue (NCT05437341). A second Phase I trial testing the feasibility, safety, and efficacy of anti-GD2/PSMA bi-specific CAR-T cell therapy in patients with GD2 and PSMA-positive tumors (including TNBC), is currently ongoing (NCT05437315). Moreover, in China, a third Phase I trial is currently verifying the feasibility, safety, and efficacy of PSMA-specific CAR-T cell therapy in patients with PSMA positive neoplasms, including TNBC (NCT04429451). To date, no results have been published yet.

### 4.21. Folate Receptor Alpha

In normal cells, the DNA synthesis pathway is efficient in the presence of folate, which is conducted inside the cell through a suitable receptor called the folate receptor (FRα). FRα is often overexpressed in BC, especially in TNBC and that correlates with poor clinical outcomes [95,109]. FRα-CAR-T cells were demonstrated to target FRα + TNBCs and to reduce tumor growth in MDA-MB-231 tumor xenograft [110]. To limit toxicity, Lanitis et al. designed a trans-signaling CAR with two different signaling domains (CD3ζ and CD28) located in two different CARs and one T cell to link with mesothelin and FRα in tumor cells. In these conditions, the activation occurs only in the case of simultaneous antigen linkage, and that in turn may activate T cell activity [111]. Therefore, FRα could become a potential target for immunotherapy in BC.

## 5. Overcoming the CAR-T- Related Problems in Solid Tumors: Macrophage-Based Cell Therapeutics

To date, CAR-T cell therapy has shown efficacy against hematological neoplasms, but conversely, its results against solid tumors were disappointing. The reason for this difference probably lies in the presence of a TME around solid tumors, which modulates the immune response against malignant cells, preventing the penetration of CAR-T cells [128,129,130]. To overcome the obstacle constituted by the TME, CAR molecules were conceived, taking inspiration from gamma-delta (γδ) T and natural killer (NK) cells for their specific biological features. Indeed, these cells can identify a wide range of tumor-associated antigens (TAAs) independently of the major histocompatibility complex (MHC) [131,132], with a consequently lower impact in terms of immune-mediated toxicity [133]. However, problems in the expansion process of these immune cells limit their application in clinical practice [134]. On the other hand, the phenomenon of “T cell exhaustion” represents a widely known and not a fully resolved topic [135,136,137]. Therefore, it was necessary to look for other strategies, for example, by exploring other immune cells [138]. In this context, macrophages seem to be an interesting option for immunotherapy development. Indeed, they present a wide range of immune activities, including their role as antigen-presenting cells which may modulate adaptive immune responses, phagocytosis, and pro-inflammatory cytokine secretion [139]. Moreover, they constitute a considerable percentage of immune cells within the TME of solid tumors, which recruit peripheral blood monocytes and then promote their differentiation into tumor-associated macrophages (TAMs) [140,141].

TAMs are often reported as both M1 (pro-inflammatory) and M2 (anti-inflammatory) phenotypes, but a higher M2 concentration is more frequently associated with a poor prognosis [142]. Their presence is considered crucial for TME regulation, especially regarding stimulation of tumor growth, angiogenesis, and metastatization [143,144]. Moreover, it was reported that TAMs have a role in cytotoxic lymphocyte recruitment in the TME [145].

In light of all of this, targeting TAMs has become the goal of numerous immunological approaches, such as TAM depletion, repolarization, or inhibition of TAM-secreted suppressive molecules [146]. Moreover, instead of directly targeting TAMs, different studies have evaluated the role of macrophages in cancer therapy. For example, antibody-dependent cellular phagocytosis is an interesting strategy (ADCP) [146]. It uses antibodies against specific tumor-associated antigens via the Fab region, which are internalized through the binding of Fc receptors (such as CD16a or CD32a) on macrophages. Moreover, macrophages may stimulate phagocytosis through these receptors and other surface molecules, including Mac1 or LRP1, whose intracellular mechanism of action is similar to that played by CD16a and CD32a. Indeed, their cytoplasmic region is rich in tyrosine-based activation motifs, which can activate MAPK and PI3K/AKT signaling pathways, with a consequential phagocytosis process against cancer cells [147]. Bispecific antibodies targeting different TAAs or macrophage receptors is another option [148]. Engaging phagocytosis checkpoint inhibitors, such as CD47, could enhance phagocytosis mechanisms by blocking the macrophage-inactivating signals [138]. However, using antibodies is not so simple; to date, some challenges must be overcome before they are clinically developed. Firstly, macrophages expose, on their extracellular membrane, the inhibitory FC receptor (FcγRIIb), which counteracts cell activation. Secondly, Mab-therapy cannot discriminate between antitumoral and/or protumoral TAMs [149].

For these reasons, another option is adoptive cell therapy based on ex vivo genetically engineered CAR macrophages (CAR-Ms). Among the different types of CAR molecules, second and third-generation CARs are preferred because of their capacity to potentiate phagocytic activation signals [150,151]. CAR-Ms provide some advantages with regards to T cells: (1) a lower risk of GVHD, which allows CAR production in advance for “on-demand” use; and (2) a significant production of MMPs, which allows macrophages to degrade ECMs and, consequently, get close to the tumor cells [152]. However, some problems are still unresolved: (1) Although their efficacy and safety profile have been reported in animal studies, in humans, it is still unclear; and (2) the use of viral transfection in CAR gene transfer could promote insertions with an unforeseeable impact on treatment. In this context, the CRISPR/Cas9 genome targeting system could represent a valuable option to overcome this problem [153].

Moreover, regenerative medicine could represent a potential strategy for limiting the high cost of CAR therapy by providing a sustainable source of CAR-Ms. Delivering CARs to induced pluripotent stem cell (iPSC)-derived macrophages may extend CAR-M cell therapy to a larger-patient population. In a recent study, iPSC-derived CAR-Ms reduced tumor growth by activating phagocytosis in leukemia, ovarian, and pancreatic cancer cell lines. Moreover, the same results were reported in vivo in an ovarian cancer mouse model [154].

## 6. CAR-M in Solid Tumors and BC

To date, several researchers have attempted to employ CAR-M against solid tumors and BC. Different CAR-phagocytes (CAR-P) have been designed to guide macrophages against specific targets. In particular, CAR-P expressing the FcRv or Megf10 intracellular region was shown to stimulate phagocytosis of TAA by the TCR-CD3ζ-mediated recruitment of SYK kinase. Usually, complete phagocytosis is uncommon, suggesting that the CAR-P macrophage link with target cells is insufficient to obtain that. In this context, it is worth remembering that the PI3K signal pathway demonstrated its involvement in target internalization and phagocytosis enhancement in macrophages [152]. For this reason, a “tandem” CAR (CAR-P tandem) has been conceived by connecting the PI3K p85 subunit with CAR-P-FcRv. This molecule demonstrated an increase in the phagocytic activity of CAR-P, especially in terms of whole-cell phagocytosis [150].

CAR-147 is a CAR molecule that is composed of a single-strand antibody fragment targeting HER2, a murine hinge region of IghG1, and a trans-membrane and intracellular domain of mice-derived CD147. Co-culturing CD147 with HER2 + human BC cells led to an intense MMP expression, demonstrating the capacity of CAR-147 to target HER2 and effectively promote MMP production in macrophages. Indeed, CAR-147 macrophages were shown to increase the amount of T cells close to tumor cells compared with those in tumors that were treated with controlled macrophages, demonstrating their potential to destroy the extracellular matrix into tumors. Moreover, CAR-147 macrophages have shown an antitumor effect by increasing IL-12 and IFNγ levels in tumor tissue [155]. An intravenous CAR-147 injection significantly inhibited cancer growth in 4T1 BC mouse models, but the same was not shown in vitro. Recently, at the University of Pennsylvania, an adenovirus-induced CAR-M composed of an anti-HER2 CAR and the CD3ζ intracellular domain was designed, demonstrating in vitro its specificity in terms of antigen-specific phagocytosis against HER2-positive tumor cells. A single injection of anti-HER2 CAR-M was shown to reduce tumor load and prolong survival in mice. It was also able to transform M2 macrophages into M1 macrophages, stimulate an inflammatory TME and promote anti-tumor cytotoxicity. In addition, HER2 CAR-M may produce epitope diffusion, which could become another solution for avoiding tumor immune escape [151,156].

Another study combined an anti-HER2 CAR with transduced primary human CD14+ peripheral blood monocyte-derived macrophages. These CAR-Ms promoted phagocytosis of the HER2+ ovarian cancer cell line SKOV3 in a dose-dependent manner. The authors further demonstrated that macrophage transduction is unaffected by the anticancer effect since their transduction with a control CAR lacked antitumor activity [151].

Moreover, in vivo, the SKOV3 tumor burden in NOD-SCID mice was considerably lower in the cases that were treated with primary human anti-HER2 CAR-Ms. The authors also demonstrated that CAR-Ms survived and resisted the immunosuppressive cytokines that were secreted by the TME. On the contrary, CAR-Ms secreted pro-inflammatory cytokines, determining a macrophage conversion from an M2 to an M1 phenotype, and consequently transforming TME into a proinflammatory environment. Furthermore, a combination of donor-derived T cells with CAR-Ms increased the antitumor response in vivo [151].

Pierini et al. demonstrated that the infusion of murine-derived anti-HER2 CAR-Ms determined an inhibition of tumor growth, a prolongation of overall survival, and an increase of CD4+ and CD8+ T cells, NK cells, and dendritic cells in the TME. The authors also reported that CAR-Ms have a critical role in regulating the TME through the upregulation of MHC I/II expression on the cancer cells [157].

## 7. Clinical Applications of CAR-M Strategy against BC and Other Solid Tumors

Until December 2022, three clinical trials have evaluated a CAR-M-based strategy in solid tumors, two of which achieved FDA approval [152] (Table 2).

The first one (a Phase I clinical trial) tested CT-0508 (CARISMA Therapeutics Inc., Philadelphia, PA, USA), a therapy consisting of anti-HER2 CAR macrophages infused in 18 patients with relapsed/refractory HER2 over-expression tumors. The study evaluated the effects of adenovirus transduction CAR-M. The estimated study completion date is February 2023 (NCT04660929). The second trial tested MCY-M11 (MaxCyte Inc., Gaithersburg, MD, USA), consisting of mRNA-targeted PBMCs (not only CAR-M) which express mesothelin-CAR, in patients with relapsed/refractory ovarian cancer and peritoneal mesothelioma (NCT03608618). A third trial (CARMA-2101), not yet recruiting, will be conducted at the Centre Oscar Lambret (Lille, France). This observational study aims to determine the antitumor activity of new CAR-Ms in 100 BC patients’ derived organoids. In particular, researchers will test the CAR-M activity against organoids that are derived from HER2-negative, HER2 low, and HER2-positive BC, and then they will compare the activity of CAR-Ms and non-modified macrophages. The estimated study completion date is 1 September 2023 (NCT05007379).

## 8. Conclusions

Recently, the remarkable advances reported in the field of immunotherapy have profoundly changed our approach toward many types of cancer. By gaining a better understanding of the role immune cells play in tumor progression mechanisms, it was possible to develop both drugs directed against specific immunological targets and forms of immune cell-based therapy, such as CAR technologies that led to the creation of CAR-T cells that are also effective in the clinical setting (especially in the field of haematological malignancies).

To date, several problems significantly limit the application of CAR-T cells especially toward solid tumors. Due to this, further studies of CAR-M in tumor therapy are interesting because of the known adaptability of these immune cells to solid tumors.

Indeed, the first results have shown that CAR-M is very promising in the fight against cancer; preclinical data have confirmed their efficacy (in terms of tumor phagocytosis and growth inhibition both in vitro and in vivo) and also in several solid tumors, including TNBC. The latter represents a heterogeneous BC subtype that is usually resistant to standard therapies. However, its immunogenic nature led to favorable clinical benefits from the new immune checkpoint inhibitors, such as atezolizumab, recently approved by the FDA in combination with nab-paclitaxel against metastatic TNBC [158].

With regards to CAR-based therapy in TNBC, this is an emerging field, whose improvement depends on discovering the suitable and targetable TAAs, mostly in preclinical and early clinical stages. In our article, we discussed different novel CAR-based target antigens evaluated against BC. A lot of them have only been tested in “in vitro” and “in vivo” studies, and a small part of them were also evaluated in humans, as summarized in Table 1 and Table 2. Furthermore, scientific research is discovering other potential targets, such as specific embryonic antigen-4, which was evaluated in “in vitro” and “in vivo” studies in BC patients, but with less data to date in comparison with the aforementioned targets [159].

Therefore, we still have little clinical data to judge the effective validity of these new therapeutic approaches against BC (and not only), which need further studies. Due to this shortage of relevant data on small series, there are still many unanswered questions. For example, is CAR-based (M or T cell) therapy more effective than standard treatments against TNBC? Is it enough to consider a mono-immunotherapy, or should it be combined with other strategies, perhaps even with other types of immunotherapy (e.g., with immune checkpoint inhibitors)? Regarding the latter question, it appears that combination regimens may lead to better efficacy, especially in terms of overcoming the TME. Indeed, in a recent study that was conducted in immunocompetent mouse models of HER2+ solid tumors (including BC), a combination of anti-PD-1 with HER2 CAR-M cell therapy demonstrated better OS and tumor control than monotherapy strategies [78]. In this regard, other strategies were evaluated. For example, ECM- or cancer-associated fibroblasts (CAF)-targeting and macrophage- or monocyte-eliminating agents were tested with the aim to enhance CAR-T antitumor effects in TNBC also [160].

It seems clear that the success of CAR-based therapy in BC will depend on the ability to select the best antigens to be used as a basis for the development of more effective, and at the same time more manageable and less toxic CAR molecules. In the near future, it is hoped that some more data can be obtained from ongoing studies.

## Figures and Tables

**Figure 1 cancers-15-01597-f001:**
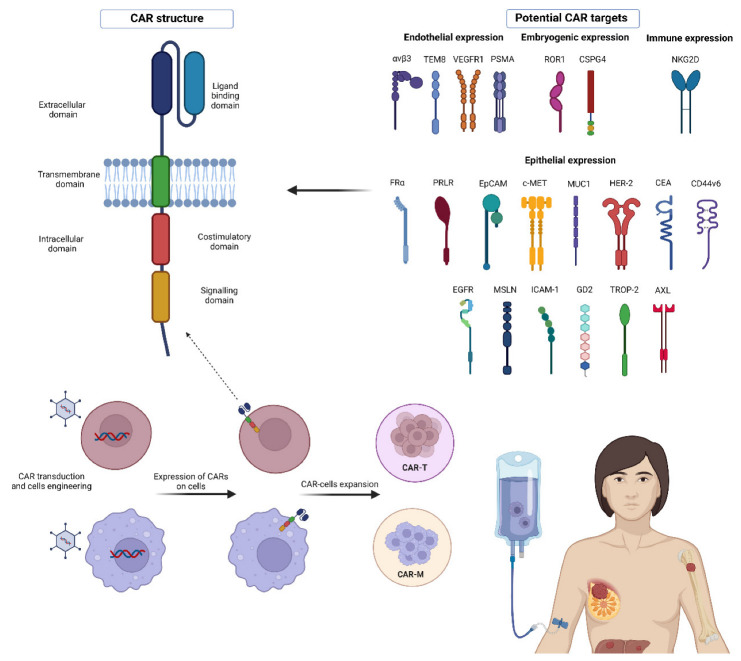
The image summarizes all the molecules that are currently being evaluated as potential targets for CAR development in BC.

**Table 1 cancers-15-01597-t001:** Trials testing the potential targets for CAR-T cell therapy in BC.

Target	Expression in Healthy Tissue	Role	Drug Agent	Trial	Status
αvβ3-integrin	Platelets, macrophages, dendritic cells, activated endothelial cells [41,42]	Cell proliferation, adhesion, metastatization, angiogenesis [41]		only preclinical data [41,42,43,44,45,46]	
TEM8	Endothelium [47,48]	Endothelial cell development [47,48]	L2CAR-T	only preclinical data [48,49,50,51]	
AXL	Bone marrow stroma and myeloid cells [52]	Tumor expansion, metastasization and survival [52]	AXL-CAR-Tcells	only preclinical data [53,54]	
CSPG4	Oligodendrocyte progenitor cells [55]	neuronal network regulation and epidermal stem cells homeostasis [55]	CSPG4- CAR-T cells	only preclinical data [56]	
TROP-2	Epithelial tissue [57]	Invasiveness [57]		only preclinical data [58]	
GD2	Neuroectoderm [59]	Cell signal transduction modulation [59]	Her2, GD2, and CD44v6	NCT04430595	ongoing
ICAM-1	Endothelial cells and immune cells [60]	Migration, invasiveness [61]		Only preclinical data [62,63]	
Mesothelin	Mesothelial cells [64,65]	Cell adhesion [64,65]	Anti-mesothelin CAR-T cells	NCT01355965 NCT02580747 NCT02414269 NCT02792114	completed unknown ongoing ongoing
MUC1	Epithelial tissue [66]	Production of mucin [67,68,69]	huMNC2-CAR44 CAR T-TnMUC1 TILs/CAR-TILs targeting multiple antigens *	NCT04020575 NCT04025216 NCT04842812	ongoing ongoing ongoing
ROR1	Embryogenic tissue [70]	Cell survival and differentiation in embryogenesis [70]	ROR1-CAR-T TILs/CAR-TILs targeting multiple antigens *	NCT02706392 NCT04842812	ongoing ongoing
EpCAM	Epithelial tissue [71]	Cell adhesion [71]	EpCAM CAR-T	NCT02915445	ongoing
EGFR	Epithelial tissue [72]	Cell survival, proliferation [72]	EGFR/B7H3 CAR-T, TGFβR-KO CAR-EGFR T Cells	NCT05341492, NCT04976218	ongoing ongoing
HER-2	Epithelial tissue (in particular breast, skin, and gastrointestinal, respiratory, reproductive, urinary tracts) [73]	Cell proliferation, differentiation, and survival [73]	CCT303-406 CAR-T BPX-603 CAR-T HER-2 CAR-T HER-2 CAR-T + CAdVEC ** TILs/CAR-TILs targeting multiple antigens *	NCT04511871 NCT04650451 NCT03696030 NCT03740256 NCT04842812	ongoing ongoing ongoing ongoing ongoing
NKG2D	Innate and adaptive immune cells [74]	Cytotoxicity and secretion of cytokines [74]	NKG2DL CAR-T cells	NCT04107142	unknown
CEA	Epithelia (in particular enteric tissue) and embryogenic tissue [75]	Cell migration, proliferation, and survival [75]	CEA CAR-T cells	NCT04348643	ongoing
CD44v6	Epithelial tissue and hematopoietic cells [76]	Cell survival, proliferation, migration [76]	4SCAR-CD44v6 T-cell	NCT04427449	ongoing
PSMA	Endothelium, prostate [77]	Angiogenesis and immune modulation [77]	bi-4SCAR PSMA/CD70 Tcells, bi-4SCAR GD2/PSMA T cells, 4SCAR-PSMA T cells	NCT05437341 NCT05437315 NCT04429451	ongoing ongoing ongoing
FRα	Epithelial tissue (mammary ducts, lungs, kidneys the choroid plexus) [78]	Cell growth and survival [79]		only preclinical data [80]	
c-MET	Epithelial tissue [81]	Cell differentiation, proliferation, migration, angiogenesis, and epithelial-mesenchymal transition [82]	cMet RNA CAR- T cells	NCT01837602	completed

Abbreviations: AXL = tyrosine-protein kinase receptor UFO; CAR-T = chimeric antigen receptor T cells; CD44v6 = CD44 variant domain 6; CEA = carcinoembryonic antigen; c-MET = tyrosine-protein kinase Met; CSPG4 = chondroitin sulfate proteoglycan-4; EGFR = epidermal growth factor receptor; EpCAM = epithelial cell adhesion molecule; FRα= folate receptor alpha; GD2 = Disialoganglioside; ICAM-1 = Intercellular Adhesion Molecule-1; MUC1 = Mucin1; NKG2D = natural killer group 2, member D ligand; PSMA = prostate-specific membrane antigen; ROR1 =Receptor tyrosine kinase-like orphan receptor; TEM8 = tumor endothelial marker; TILs = tumor-infiltrating lymphocytes; TROP-2 = trophoblast cell-surface antigen 2. * TILs and CAR-TILs targeting HER2, Mesothelin, PSCA, MUC1, Lewis-Y, GPC3, AXL, EGFR, Claudin18.2/6, ROR1, GD1, or B7-H3. ** CAdVEC is an oncolytic adenovirus designed to help the immune system, including HER2-specific CAR-T cells, react to the tumor.

**Table 2 cancers-15-01597-t002:** Trials on CAR-M in BC and other solid tumors.

Target	Drug Agent	Trial	Status
HER-2	CT-0508	NCT04660929	Recruiting
Mesothelin	Intraperitoneal MCY-M11 and cyclophosphamide	NCT03608618	Terminated (no results posted yet)
HER-2	HER-2 CAR-M	NCT05007379	Not yet recruiting

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
