# Peer review of "The New Frontier of Immunotherapy: Chimeric Antigen Receptor T (CAR-T) Cell and Macrophage (CAR-M) Therapy against Breast Cancer"

_cancers, 2023, doi:10.3390/cancers15051597_

Round 1
Reviewer 1 Report
Nice review of the topic. Please revise for minor corrections.
For example: "Tumor Endothelial Marker (TEM)8, "vs TEMP8 in line 164
Author Response
Comments and Suggestions for Authors
Nice review of the topic. Please revise for minor corrections. For example: "Tumor Endothelial Marker (TEM)8, "vs TEMP8 in line 164. RE: We are very grateful to the reviewer for his comments. We've changed the correction as requested.
Reviewer 2 Report
Dear Authors,
first of all, congratulations for your effort and providing an interesting viewpoint to the immunotherapy landscape. I hope my comments will help you to improve your paper. English correction is also required. Comas, spaces and punctuation in general should be improved as well. Especially spaces are doubled in many places of the text.
line 18 --> immune checkpoint inhibitors, instead of immunocheckpoint
36 --> TNBC definition to be improved, why do we call it "triple"?
46 --> neoantigen term should be explained
65 --> many punctation mistakes makes it incomprehensible, eg. CD8+ T cells instead of CD8T+ or similar; this applies to entire manuscript as it looks like written by someone with dyslexia maybe; typos of this kind are inadmissible in science
73 --> aBC, a is what?
81 --> TILs and T cells: T cells compose TILs, so what's the difference here?
Fig.1. --> the idea is very good, however the picture requires some amendments in order to be fully understood, for example arrows are hidden and confusing; also, some inscriptions would be welcome for example on cell types
Table 1. Are those all trials ongoing currently or only selected trials? If selected, what was the basis of selection?
252 --> extremely interesting and important information, maybe there are more trials like this one, or at least experiment results indicating such potential targets?
316 --> not clear why folate receptor is a good target for immunotherapy. It is present in many cells, thus, toxicity may be quite significant.
395 --> PI3K pathway is mentioned here, but not everyone may know why, what is the connection with the previously described topic?
477 --> the combination therapy mentioned in the last sentence is very interesting, maybe it will be a nice idea to expand it?
Overall, the topic is very important and interesting. However, abbreviations and scientific names are not always used in a proper way or written with mistakes, which may result from poor English performance. Language correction will be crucial.
Author Response
Comments and Suggestions for Authors
Dear Authors,
first of all, congratulations for your effort and providing an interesting viewpoint to the immunotherapy landscape. I hope my comments will help you to improve your paper. English correction is also required. Comas, spaces and punctuation in general should be improved as well. Especially spaces are doubled in many places of the text. RE: We thank the reviewer for his suggestions, which were very useful for the improvement of the manuscript. We have corrected spelling and punctuation errors as you will see in the manuscript. We also arranged for the article to be reviewed by a native English speaker
line 18 --> immune checkpoint inhibitors, instead of immunocheckpoint. RE: modified as requested
36 --> TNBC definition to be improved, why do we call it "triple"? RE: We thank the reviewer for his suggestion. Our manuscript is aimed primarily at oncologists, so we have only hinted at explaining commonly used terms. Anyway, two lines later we specified it “TNBC is defined by the lack of expression of HR and HER-2 and accounts for approximately 15% of all BCs”
46 --> neoantigen term should be explained RE: We thank the reviewer for his suggestion. Our manuscript is aimed primarily at oncologists, so we have only hinted at explaining commonly used terms. Anyway, in the same sentence we mentioned that this is an immunocheckpoint.
65 --> many punctation mistakes makes it incomprehensible, eg. CD8+ T cells instead of CD8T+ or similar; this applies to entire manuscript as it looks like written by someone with dyslexia maybe; typos of this kind are inadmissible in science. RE: We thank the reviewer for his suggestion. We modified as requested
73 --> aBC, a is what? RE: We thank the reviewer for his suggestion: “a” means “advanced”; we modified by writing “advanced” instead of “a”
81 --> TILs and T cells: T cells compose TILs, so what's the difference here? RE: TILs are not only composed of T cells, but also of other cells that can be used to produce CAR, such as NK cells or macrophages, so TILs and T cells are not the same thing.
Fig.1. --> the idea is very good, however the picture requires some amendments in order to be fully understood, for example arrows are hidden and confusing; also, some inscriptions would be welcome for example on cell types. RE: we are very grateful to the reviewer for his comment. We modified the image based on his suggestions
Table 1. Are those all trials ongoing currently or only selected trials? If selected, what was the basis of selection? RE: Those are all trials currently underway
252 --> extremely interesting and important information, maybe there are more trials like this one, or at least experiment results indicating such potential targets? RE: we are very grateful to the reviewer for his comment. To date, the reported reference is the only one dealing with a bispecific antibody based on p95HER2
316 --> not clear why folate receptor is a good target for immunotherapy. It is present in many cells, thus, toxicity may be quite significant. RE: We are very grateful to the reviewer for this comment. Indeed, the ideal target should be expressed only by the tumor cell, but this is not always possible. In this case, its use as a potential target is determined by its overexpression in TNBCs, as reported in the text.
395 --> PI3K pathway is mentioned here, but not everyone may know why, what is the connection with the previously described topic? RE: We are very grateful to the reviewer for this comment. The role of the PI3K pathway in the stimulation of macrophage phagocytic activity has only been mentioned in the text, not being the subject of the review. As for the phagocytic activity of macrophages, we discussed it a little more extensively in the previous paragraph
477 --> the combination therapy mentioned in the last sentence is very interesting, maybe it will be a nice idea to expand it? RE: We are very grateful to the reviewer for this comment. These are the final comments, but there is not much data, however we have inserted the following sentence: “In the near future, it is hoped that some more data can be obtained from ongoing studies”.
Overall, the topic is very important and interesting. However, abbreviations and scientific names are not always used in a proper way or written with mistakes, which may result from poor English performance. Language correction will be crucial. RE: We thank the reviewer for his suggestions, which were very useful for the improvement of the manuscript. We have corrected spelling and punctuation errors as you will see in the manuscript. We also arranged for the article to be reviewed by a native English speaker
Reviewer 3 Report
The authors mainly discussed the current status of CAR-T cell and CAR-M therapies against breast cancer. The authors also pointed out in the manuscript that CAR-T cell and CAR-M therapies on solid tumor achieved limited success, whether they are superior to other immunotherapies are yet to be discovered.
In general, the manuscript covered the topic of CAR-T cell and CAR-M therapies against breast cancer well. In the introduction, the authors mentioned the current status of immunotherapy against breast cancer and background information about breast cancer. However, later in the manuscript, the author mainly focused on CAR-T and CAR-M therapies. There is not much discussion about CAR-T and CAR-M therapies and other immunotherapies against different types of breast cancer. Giving a brief overview of current other immunotherapies against breast cancer can give readers more insights into the pros and cons of other therapies compared to CAR-T and CAR-M.
The author summarized current CAR-T cell generation, CAR construct, and current targets. But with only word description, it may be a little hard for readers from other fields to understand. It would help readers to comprehend the concepts if such information is included in the illustration. The manuscript summarized current CAR-T and CAR-M therapies against breast cancer well and can really offer the community a clear overview of them.
In the discussion part, the author would like to point out that current researches on CAR-T and CAR-M against breast cancer are limited and hard to draw conclusions about them. It would be much better if the author can discuss breast cancer with other solid tumors, and point out some potential improvement directions of CAR-T and CAR-M research against breast cancer specifically.
The language of the manuscript is professional and well-written. But there are still some formatting issues. For example, the space between two words sometimes varies within a line (Eg. Line 48), and extra spaces after symbols (Eg. Line 65 after “[”). Figure 1 can be improved by supplementing the figure about where those targets are expressed in the tumor.
In summary, this manuscript is a good reference for onco-immunology community. The review focuses on CAR-T and CAR-M therapy researches against breast cancer and current clinical trials. The statements and conclusions are drawn properly and fairly.
Author Response
Comments and Suggestions for Authors
The authors mainly discussed the current status of CAR-T cell and CAR-M therapies against breast cancer. The authors also pointed out in the manuscript that CAR-T cell and CAR-M therapies on solid tumor achieved limited success, whether they are superior to other immunotherapies are yet to be discovered. In general, the manuscript covered the topic of CAR-T cell and CAR-M therapies against breast cancer well. In the introduction, the authors mentioned the current status of immunotherapy against breast cancer and background information about breast cancer. However, later in the manuscript, the author mainly focused on CAR-T and CAR-M therapies. There is not much discussion about CAR-T and CAR-M therapies and other immunotherapies against different types of breast cancer. Giving a brief overview of current other immunotherapies against breast cancer can give readers more insights into the pros and cons of other therapies compared to CAR-T and CAR-M. RE: we are very grateful to the reviewer for his comment and agree with that idea, which may be worthy of a separate manuscript. However, immunotherapy in breast cancer in general represents too broad a topic, which is beyond the scope of our article
The author summarized current CAR-T cell generation, CAR construct, and current targets. But with only word description, it may be a little hard for readers from other fields to understand. It would help readers to comprehend the concepts if such information is included in the illustration. The manuscript summarized current CAR-T and CAR-M therapies against breast cancer well and can really offer the community a clear overview of them. RE: we are very grateful to the reviewer for his comment. We modified the image based on his suggestions
In the discussion part, the author would like to point out that current researches on CAR-T and CAR-M against breast cancer are limited and hard to draw conclusions about them. It would be much better if the author can discuss breast cancer with other solid tumors, and point out some potential improvement directions of CAR-T and CAR-M research against breast cancer specifically.
RE: We thank the reviewer very much for his interesting comments. Unfortunately, there are no characteristics of breast cancer that can determine a clear difference regarding the use of CAR-T cell therapy compared to other solid tumors; we have dealt with TNBC because it is the most suitable histological type to respond to this type of therapy, but to date we have no further data in this sense; moreover, the data of the studies available to date are not specific for breast cancer, but they are often tumors selected on the basis of molecular expression rather than histology; instead, as regards the improvements in research on CAR molecules, we have dealt with in paragraph 5 "Overcoming the CAR-T- related problems in solid tumors: Macrophages-based cell therapeutics"
The language of the manuscript is professional and well-written. But there are still some formatting issues. For example, the space between two words sometimes varies within a line (Eg. Line 48), and extra spaces after symbols (Eg. Line 65 after “[”). Figure 1 can be improved by supplementing the figure about where those targets are expressed in the tumor. RE: We thank the reviewer for his suggestions, which were very useful for the improvement of the manuscript. We have corrected spelling and punctuation errors as you will see in the manuscript. We also arranged for the article to be reviewed by a native English speaker
In summary, this manuscript is a good reference for onco-immunology community. The review focuses on CAR-T and CAR-M therapy researches against breast cancer and current clinical trials. The statements and conclusions are drawn properly and fairly.
RE: We thank the reviewer for his suggestions, which were very useful for the improvement of the manuscript.
Reviewer 4 Report
The topic is interesting and highly relevant at present. Given the current development of adoptive T cell therapy in solid tumors, it is worthwhile to publish a review on this topic.
However, this review needs a major revision
Title, Summeray and main text should have the same focus, in the main text there is also a focus on bispecifics, in the title then maybe better T cell directed therapy againtst breast cancer?
Section 1
Since bispecifics are listed several times under 4. they should also be mentioned under 1.
Section 4
List expression in healthy tissue as well information of the trials in the table.
A listing of all targets with corresponding trials and their completions dates in the main text is difficult to read. It would make more sense to have a table in which all targets are listed with the corresponding trials including the expression of the targets in healthy tissue and thereie is an overall discussion on targets in the main text. It might be also useful to rank currently investigated targets. Is there an ideal tumor antigen? There should also be comments on potential on target toxicity.
Furthermore, the authours should comment in an extra section on the lymphodepletion regimen (comparison of dose and agents with hematologic neoplasms), possible administration modalities (Iv, different local application options, etc) as well as single vs multipe adoptive cell transfers (T cells, NK cells, macrophages).
Page 7, line 263: VEGFR1 represents a potential candidate for bispecific antibody therapy ... why only for bispecific and not for CAR T ?
Page 8, line 287 mention the histiotypes
Page 8, line 313/ 314 : Please be more specific here. Please comment also on whether, CD44V6 is expressed in healthy cells .
Page 8, line 317: Please comment on current development of drugs specific for Folate receptor.
Section 5
Page 9, line 343: Into TAMS are often reported ... ?
line 346 / 347 : please be more specific
line 376/ 377 this applies also to CAR / NK T cells
line 436: I guess you mean not drugs that achieved FDA approval but the trials?
Conclusion: also comment shortly on potential targets and applications routs
Minor: check typing
Author Response
Comments and Suggestions for Authors
The topic is interesting and highly relevant at present. Given the current development of adoptive T cell therapy in solid tumors, it is worthwhile to publish a review on this topic. However, this review needs a major revision. Title, Summery and main text should have the same focus, in the main text there is also a focus on bispecifics, in the title then maybe better T cell directed therapy againtst breast cancer?
RE: we thank the reviewer very much for the suggestion; however, the article deals not only with T cell-based immunotherapy but also with CAR-M. In paragraph 4 we also dealt with bispecifics, but then the CAR-M strategy represents a further step forward.
Section 1
Since bispecifics are listed several times under 4. they should also be mentioned under 1.
RE: we thank the reviewer very much for the suggestion. In paragraph 1, we mentioned Bispecific abntibodies as follows: “New emerging treatments in solid tumors are also involving immunotherapy after the incredible results achieved in the onco-hematology field. Among them, Bispecific antibody therapy and Adoptive cell therapy (ACT); the latter exploits TILs or T cells genetically engineered to express modified T-cell receptors (TCR) or chimeric antigen receptors (CAR).”
Section 4
List expression in healthy tissue as well information of the trials in the table. A listing of all targets with corresponding trials and their completions dates in the main text is difficult to read. It would make more sense to have a table in which all targets are listed with the corresponding trials including the expression of the targets in healthy tissue and thereie is an overall discussion on targets in the main text. It might be also useful to rank currently investigated targets. Is there an ideal tumor antigen? There should also be comments on potential on target toxicity.
RE: we are very grateful to the reviewer for his comment and agree with that idea, which may be worthy of a separate manuscript. However, we think that the comparison between healthy and tumor tissue in general represents too broad a topic, which is beyond the scope of our article. However, in figure 1, we separated the potential targets according with their localization.
Furthermore, the authours should comment in an extra section on the lymphodepletion regimen (comparison of dose and agents with hematologic neoplasms), possible administration modalities (Iv, different local application options, etc) as well as single vs multipe adoptive cell transfers (T cells, NK cells, macrophages).
RE: we are very grateful to the reviewer for his comment and agree with that idea, which may be worthy of a separate manuscript. However, we think that the comparison between haematologial and solid tumors in general represents too broad a topic, which is beyond the scope of our article
Page 7, line 263: VEGFR1 represents a potential candidate for bispecific antibody therapy ... why only for bispecific and not for CAR T ? RE: We thank the reviewer fori his suggestion. We modified as follows: “This Tyrosin-kinase receptor is involved in migration and survival of hematopoietic stem cells, and its overexpression is related with the process of BC metastatization [68,69]. Therefore, VEGFR1 represent a potential candidate for Immunotherapy.”
Page 8, line 287 mention the histotypes RE: ok, we modified as follows: “analyzing tumor specimens (4 TNBC and 2 ER+ HER2 negative BCs)”
Page 8, line 313/ 314 : Please be more specific here. Please comment also on whether, CD44V6 is expressed in healthy cells RE: ok we added a specific sentence as fopllows: “In normal tissue, its presence is reported only on epithelial and hematopoietic cell subgroups, especially during embryogenesis and hematopoiesis”.
Page 8, line 317: Please comment on current development of drugs specific for Folate receptor. RE: OK, we modified as follows “FRα is often overexpressed in BC especially in TNBC and that correlates with poor clinical outcomes [51; + Ginter PS, McIntire PJ, Cui X, Irshaid L, Liu Y, Chen Z, et al. Folate Recep-tor Alpha Expression Is Associated With Increased Risk of Recurrence in Triple-negative Breast Cancer. Clin Breast Cancer. 2017; 17: 544-9]. In fact, FRα-CAR-T cells demonstrated to target FRα+TNBCs and to reduce tumor growth in MDA-MB-231 tumor xenograft [Song DG, Ye Q, Poussin M, Chacon JA, Figini M, Powell DJ, Jr. Effective adoptive immunother-apy of triple-negative breast cancer by folate receptor-alpha redirected CAR T cells is in-fluenced by surface antigen expression level. J Hematol Oncol. 2016; 9: 56]. In order to lim-it toxicity, Lanitis et al. designed a trans-signaling CAR with two different signaling do-mains (CD3ζ and CD28) located in two different CARs and in one T cell in order to link with mesothelin and FRα in tumor cells. In this conditions, the activation occur only in case of simultaneous antigen linkage, and that in turn may activate T cell activity [Lanitis E, Poussin M, Klattenhoff AW, Song D, Sandaltzopoulos R, June CH, et al. Chimeric anti-gen receptor T Cells with dissociated signaling domains exhibit focused antitumor activi-ty with reduced potential for toxicity in vivo. Cancer Immunol Res. 2013; 1: 43-53]. Therefore, FRα could become a potential target for Immunotherapy in BC”.
Section 5
Page 9, line 343: Into TAMS are often reported ... ? RE: ok we modified that sentence as follows: “TAMs are often reported as both M1 (pro-inflammatory) and M2 (anti-inflammatory) phenotypes, but higher M2 concentration is more frequently associated with a poor prognosis”
line 346 / 347 : please be more specific RE: ok we modified the previous sentence (“TAMs are often reported as both M1 (pro-inflammatory) and M2 (anti-inflammatory) phenotypes, but higher M2 concentration is more frequently associated with a poor prognosis”), in order to clarify this part also.
line 376/ 377 this applies also to CAR / NK T cells RE: We agree, indeed in that paragraph we dealt with the CAR-T- related problems
line 436: I guess you mean not drugs that achieved FDA approval but the trials? RE: we thank the reviewer for his suggestion, we modified as requested
Conclusion: also comment shortly on potential targets and applications routs. RE: We thank the reviewer for his suggestion. However, being a topic on which there is still little data explored, we preferred to conclude with questions rather than statements.
Minor: check typing. RE: We thank the reviewer for his suggestions, which were very useful for the improvement of the manuscript. We have corrected spelling and punctuation errors as you will see in the manuscript. We also arranged for the article to be reviewed by a native English speaker
Reviewer 5 Report
The Review is well done and comprehensive. The grammar could be improved by employing an Editor specialized in English grammar. The Figure is simplistic and could be given a more artistic makeover.
Author Response
The Review is well done and comprehensive. RE: We are very grateful to the reviewer for his comments.
The grammar could be improved by employing an Editor specialized in English grammar. RE: We have corrected spelling and punctuation errors as you will see in the manuscript. We also arranged for the article to be reviewed by a native English speaker
The Figure is simplistic and could be given a more artistic makeover. RE: We modified the image based on his suggestions